# PUSHING BOUNDARIES:
# MIXUP'S INFLUENCE ON NEURAL COLLAPSE

**Quinn LeBlanc Fisher**\*, **Haoming Meng**\*, **Vardan Papyan**
University of Toronto

## ABSTRACT

Mixup is a data augmentation strategy that employs convex combinations of training instances and their respective labels to improve the robustness and calibration of deep neural networks. Despite its widespread adoption, the nuanced mechanisms that underpin its success are not entirely understood. The observed phenomenon of Neural Collapse, where the last-layer activations and classifier of deep networks converge to a simplex equiangular tight frame (ETF), provides a compelling motivation to explore whether mixup induces alternative geometric configurations and whether those could explain its success. In this study, we delve into the last-layer activations of training data for deep networks subjected to mixup, aiming to uncover insights into its operational efficacy. Our investigation (code), spanning various architectures and dataset pairs, reveals that mixup's last-layer activations predominantly converge to a distinctive configuration different than one might expect. In this configuration, activations from mixed-up examples of identical classes align with the classifier, while those from different classes delineate channels along the decision boundary. These findings are unexpected, as mixed-up features are not simple convex combinations of feature class means (as one might get, for example, by training mixup with the mean squared error loss). By analyzing this distinctive geometric configuration, we elucidate the mechanisms by which mixup enhances model calibration. To further validate our empirical observations, we conduct a theoretical analysis under the assumption of an unconstrained features model, utilizing the mixup loss. Through this, we characterize and derive the optimal last-layer features under the assumption that the classifier forms a simplex ETF.

## 1 INTRODUCTION

Consider a classification problem characterized by an input space $\mathcal{X} = \mathbb{R}^D$ and an output space $\mathcal{Y} := \{0, 1\}^C$. Given a training set $\{(\mathbf{x}_i, \mathbf{y}_i)\}_{i=1}^N$, with $\mathbf{x}_i \in \mathcal{X}$ denoting the $i$-th input data point and $\mathbf{y}_i \in \mathcal{Y}$ representing the corresponding label, the goal is to train a model $f_\theta : \mathcal{X} \mapsto \mathcal{Y}$ by finding parameters $\theta$ that minimize the cross-entropy loss $\mathrm{CE}(f_\theta(\mathbf{x}_i), \mathbf{y}_i)$ incurred by the model's prediction $f_\theta(\mathbf{x}_i)$ relative to the true target $\mathbf{y}_i$, averaged over the training set, $\frac{1}{N} \sum_{i=1}^N \mathrm{CE}(f_\theta(\mathbf{x}_i), \mathbf{y}_i)$.

Papyan, Han, and Donoho (2020) observed that optimizing this loss leads to a phenomenon called Neural Collapse, where the last-layer activations and classifiers of the network converge to the geometric configuration of a simplex equiangular tight frame (ETF). This phenomenon reflects the natural tendency of the networks to organize the representations of different classes such that each class's representations and classifiers become aligned, equinorm, and equiangularly spaced, providing optimal separation in the feature space. Understanding Neural Collapse is challenging due to the complex structure and inherent non-linearity of neural networks. Motivated by the expressivity of overparametrized models, the *unconstrained features model* (Mixon et al., 2020) and the *layer-peeled model* (Fang et al., 2021) have been introduced to study Neural Collapse theoretically. These mathematical models treat the last-layer features as free optimization variables along with the classifier weights, abstracting away the intricacies of the deep neural network.

---

\*Equal contribution

## 1.1 MIXUP

Mixup, a data augmentation strategy proposed by Zhang et al. (2017), generates new training examples through convex combinations of existing data points and labels:

$$\mathbf{x}_{ii'}^{\lambda} = \lambda \mathbf{x}_i + (1 - \lambda)\mathbf{x}_{i'}, \quad \mathbf{y}_{ii'}^{\lambda} = \lambda \mathbf{y}_i + (1 - \lambda)\mathbf{y}_{i'},$$

where $\lambda \in [0, 1]$ is a randomly sampled value from a predetermined distribution $D_\lambda$. Conventionally, this distribution is a symmetric $\text{Beta}(\alpha, \alpha)$ distribution, with $\alpha = 1$ frequently set as the default. The loss associated with mixup can be mathematically represented as:

$$\mathbb{E}_{\lambda \sim D_\lambda} \frac{1}{N^2} \sum_{i=1}^{N} \sum_{i'=1}^{N} \text{CE}(f_\theta(\mathbf{x}_{ii'}^{\lambda}), \mathbf{y}_{ii'}^{\lambda}). \tag{1}$$

A specific mixup data point, represented as $\mathbf{x}_{ii'}^{\lambda}$, is categorized as a *same-class* mixup point when $\mathbf{y}_i = \mathbf{y}_{i'}$, and classified as *different-class* when $\mathbf{y}_i \neq \mathbf{y}_{i'}$.

## 1.2 PROBLEM STATEMENT

Despite the widespread use and demonstrated efficacy of the mixup data augmentation strategy in enhancing the generalization and calibration of deep neural networks, its underlying operational mechanisms remain not well understood. The emergence of Neural Collapse prompts the following question:

> *Does mixup induce its own distinct configurations in last-layer activations, differing from traditional Neural Collapse? If so, does the configuration contribute to the method's success?*

This study aims to uncover the potential geometric configurations in the last-layer activations resulting from mixup and to determine whether these configurations can offer insights into its success.

## 1.3 CONTRIBUTIONS

Our contributions in this paper are twofold.

**Empirical Study and Discovery**   We conduct an extensive empirical study focusing on the last-layer activations of mixup training data. Our study reveals that mixup induces a geometric configuration of last-layer activations across various datasets and models. This configuration is characterized by distinct behaviours:

- **Same-Class Activations:** These form a simplex ETF, aligning with their respective classifier.
- **Different-Class Activations:** These form channels along the decision boundary of the classifiers, exhibiting interesting behaviors: Data points with a mixup coefficient, $\lambda$, closer to 0.5 are located nearer to the middle of the channels. The density of different-class mixup points increases as $\lambda$ approaches 0.5, indicating a collapsing behaviour towards the channels.

We investigate how this configuration varies under different training settings and the layer-wise trajectory the features take to arrive at the configuration. Additionally, the configuration offers insight into mixup's success. Specifically, we measure the calibration induced by mixup and present an explanation for why the configuration leads to increased calibration.

Motivated by our theoretical analysis, we also examine the configuration of the last-layer activations obtained through training with mixup while fixing the classifier as a simplex ETF.

**Theoretical Analysis**   We provide a theoretical analysis of the discovered phenomenon, utilizing an adapted *unconstrained features model* tailored for the mixup training objective. Assuming the classifier forms a simplex ETF under optimality, we theoretically characterize the optimal last-layer activations for all class pairs and for every $\lambda \in [0, 1]$.

## 1.4 Results Summary

The results of our extensive empirical investigation are presented in Figures 1, 3, 5, 10, and 12. These figures collectively illustrate a consistent identification of a unique last-layer configuration induced by mixup, observed across a diverse range of:

**Architectures:** Our study incorporated the WideResNet-40-10 (Zagoruyko & Komodakis, 2017) and ViT-B (Dosovitskiy et al., 2021) architectures;

**Datasets:** The datasets employed included FashionMNIST (Xiao et al., 2017), CIFAR10, and CIFAR100 (Krizhevsky & Hinton, 2009);

**Optimizers:** We used stochastic gradient descent (SGD), Adam (Kingma & Ba, 2017), and AdamW (Loshchilov & Hutter, 2017) as optimizers.

The networks trained showed good generalization performance and calibration, as substantiated by the data presented in Tables 1 and 2. That is, the values are comparable to those found in other papers (Zhang et al., 2017; Thulasidasan et al., 2020).

Beyond our principal observations, we conducted a counterfactual experiment, the results of which are depicted in Figures 2 and 8. These reveal a notable divergence in the configuration of the last-layer features when mixup is not employed. They also show that for MSE loss, the last-layer activations are convex combinations of the classifiers, which one may expect.

Furthermore, we juxtaposed the findings from our empirical investigation with theoretically optimal features, which were derived from an unconstrained features model and are showcased in Figure 6.

To complement these results, we train models using mixup while fixing the classifier as a simplex ETF, and we plot the last-layer features in Figure 7. This yields last-layer features that align more closely with the theoretical features.

## 2 Experiments

### 2.1 Model Training

Table 1: Test accuracy for experiments in Figures 1 and 8.

| Network | Dataset | Baseline | Mixup |
|---|---|---|---|
| | FashionMNIST | 95.10 | 94.21 |
| WideResNet-40-10 | CIFAR10 | 96.2 | 97.30 |
| | CIFAR100 | 80.03 | 81.42 |
| | FashionMNIST | 93.71 | 94.24 |
| ViT-B/4 | CIFAR10 | 86.92 | 92.56 |
| | CIFAR100 | 59.95 | 69.83 |

We consider FashionMNIST (Xiao et al., 2017), CIFAR10, and CIFAR100 (Krizhevsky & Hinton, 2009) datasets. Unless otherwise indicated, for all experiments using mixup augmentation, $\alpha=1$ was used, meaning $\lambda$ was sampled uniformly between 0 and 1. Each dataset is trained on both a Vision Transformer (Dosovitskiy et al., 2021), and a wide residual network (Zagoruyko & Komodakis, 2017). For each network and dataset combination, the experiment with the highest test accuracy is repeated without mixup and is referred to as the "baseline" result. No dropout was used in any experiments. Hyperparameter details are outlined in Appendix B.1.

### 2.2 Visualizations of last-layer activations

For each dataset and network pair, we visualize the last-layer activations for a subset of the training dataset consisting of three randomly selected classes. After obtaining the last-layer activations, they undergo a two-step projection: first onto the classifier for the subset of three classes, then onto a two-dimensional representation of a three-dimensional simplex ETF. A more detailed explanation of the projection can be found in Appendix B.2. The results of this experiment can be seen in Figure 1. Notably, activations from mixed-up examples of the same classes closely align with a simplex ETF structure, whereas those from different classes delineate channels along the decision boundary. Additionally, in certain plots, activations from mixed-up examples of different classes become increasingly sparse as $\lambda$ approaches 0 and 1. This suggests a clustering of activations towards the channels. When generating plots, we keep the network in train mode [1]. Since ViT does not have batch normalization layers, this difference is not applicable.

---

[1] We choose to have the network in evaluation mode for the WideResNet-40-10 CIFAR100 combination because the batch statistics are highly skewed due to the high number of classes.

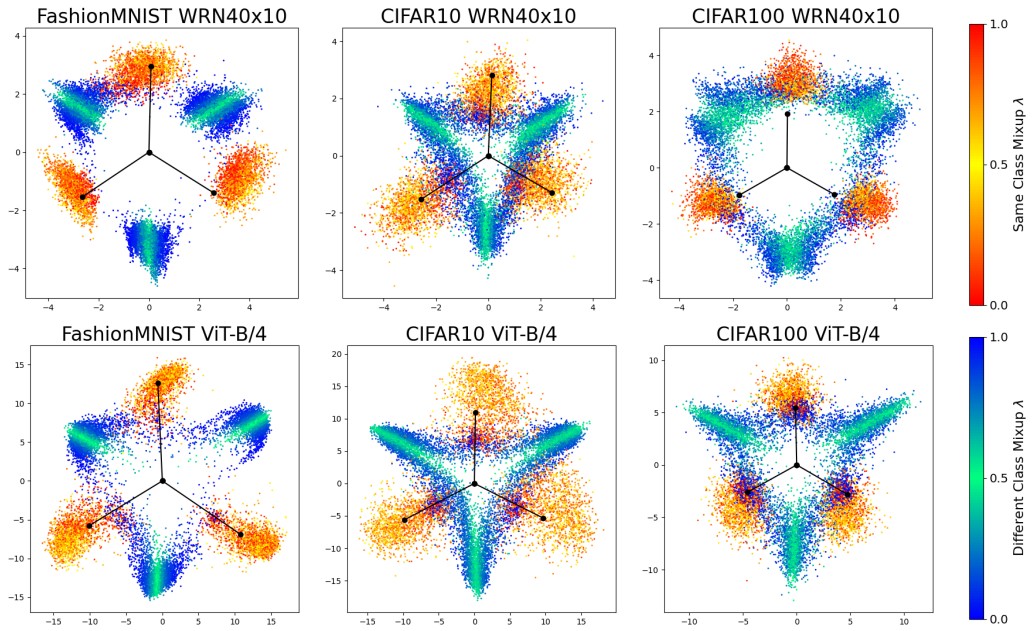

Figure 1: **(Visualization of activations outputted by networks trained with mixup)**. Last-layer activations of mixup training data for a randomly selected subset of three classes across various dataset and network architecture combinations trained with mixup. The first row illustrates activations generated by a WideResNet, while the second row showcases activations from a ViT. Each column corresponds to a different dataset. Coloration indicates the type of mixup (same or different class), along with the level of mixup, $\lambda$. For each plot, the relevant classifiers are plotted in black.

## 2.3 COMPARISON OF DIFFERENT LOSS FUNCTIONS

As part of our empirical investigation, we have conducted experiments utilizing Mean Squared Error (MSE) loss instead of cross-entropy, through which we observed in Figure 2 that features of mixed up examples are derived from simple convex combinations of same-class features. Initially, we anticipated a similar uninteresting configuration for cross-entropy; however, our measurements reveal that the resulting geometric configurations are markedly more interesting and complex. Additionally, we compare results in Figure 1 to the baseline (trained without mixup) cross-entropy loss in Figure 2. For the baseline networks, mixup data is loosely aligned with the classifier, regardless of same-class or different-class. The area in between classifiers is noisy and filled with examples where $\lambda$ is close to 0.5. Additional baseline last-layer activations can be found in Figure 8.

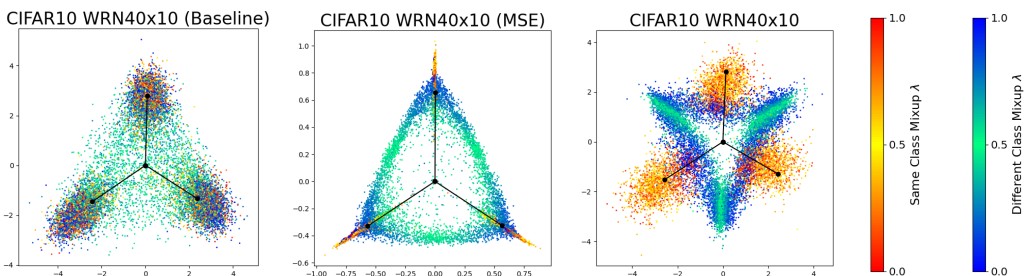

Figure 2: **(Visualization of activations outputted by networks trained with various loss functions)**. Last-layer activations for WideResNet-40-10 trained on the CIFAR10 dataset, subsetted to three randomly selected classes. Projections are generated using the same method as Figure 1. **Left to right:** baseline cross-entropy, MSE mixup, cross-entropy mixup. Colouring indicates mixup type (same-class or different-class), and the level of mixup, $\lambda$. Relevant classifiers plotted in black. Additional dataset architecture combinations for baseline cross-entropy are available in appendix D.

## 2.4 LAYER-WISE TRAJECTORY OF CLS TOKEN

Using the same projection method as in Figure 1, we investigate the trajectory of the CLS token for ViT models. First, we randomly select two CIFAR10 training images. Then, we create a selection of mixed up examples using the respective images. For each mixed up example, we project the path of the CLS token at each layer of the ViT-B/4 network.

Figure 3 presents the results for two images of the same class, and two images of a different class. For different-class mixup, the plot shows that for very small $\lambda$, the network first classifies the image as class 1 and only in deeper layers it realizes the image is also partially class 2. Furthermore, the different-class activations in preceding layers appear to be convex combinations of the unmixed activations, akin to their inputs.

The results in Figure 1 suggest that applying mixup to input data enforces a particularly rigid geometric structure on the last-layer activations. Manifold mixup (Verma et al., 2019), a subsequent technique, proposes the mixing of features across various layers of a network. The results in Figure 3 suggest that using regular mixup promotes manifold mixup-like behaviour in previous layers.

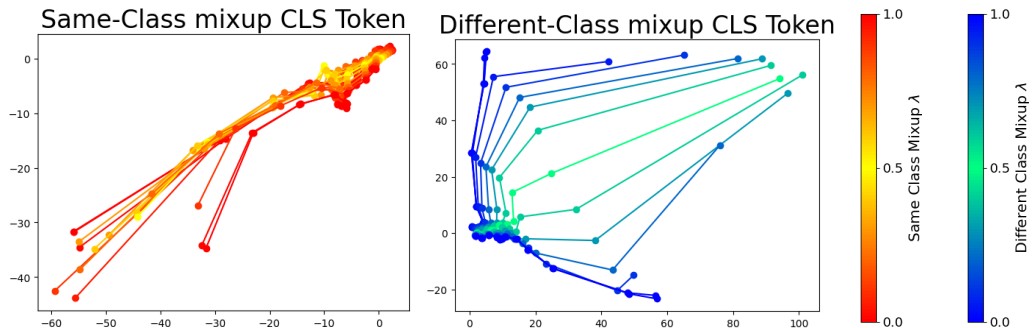

Figure 3: **(Projection of CLS token at each layer).** Projections of the CLS token of the mix up of two randomly selected training images for various values of $\lambda$. Trajectories start at the origin. Colouring indicates mixup type (same-class or different-class), and the level of mixup, $\lambda$.

## 2.5 CALIBRATION

Thulasidasan et al. (2020) demonstrated that mixup improves calibration for networks. That is, training with mixup causes the softmax probabilities to be in closer alignment with the true probabilities of misclassification. To measure a network's calibration, we use the expected calibration error (ECE) as proposed by Pakdaman Naeini et al. (2015). The exact definition of ECE can be found in Appendix C. Results for ECE can be found in Table 2. Last-layer activation plots for $\alpha = 0.4$ are available in Figure 9 in Appendix D.

Table 2: CIFAR10 expected calibration error.

| Network | Baseline | Mixup ($\alpha = 1.0$) | Mixup ($\alpha = 0.4$) |
|---|---|---|---|
| WideResnet-40-10 | 0.024 | 0.077 | 0.013 |
| ViT-B/4 | 0.122 | 0.014 | 0.019 |

The configuration presented in Figure 1 sheds light on why mixup improves calibration. Recall, mixup promotes alignment of the model's softmax probabilities for the training example $\mathbf{x}_{ii'}^\lambda$ with its label $\lambda \mathbf{y}_i + (1 - \lambda)\mathbf{y}_{i'}$. Here, $\lambda$ acts as a gauge for these softmax probabilities, essentially reflecting the model's confidence in its predictions. Turning to Figure 4, it becomes therefore evident that as $\lambda$ nears 0.5, the model's certainty in its predictions diminishes. This reduction in confidence is manifested geometrically through the spatial distribution of features along the channel. This, in turn, causes an increase in misclassification rates, due to a greater chance of activations erroneously crossing the decision boundary. This simultaneous reduction in confidence and classification accuracy leads to enhanced calibration in the model and is purely attributed to the geometric structure to which the model converged. The above logic holds as we traverse the train mixed up features but we expect some test features to be noisy perturbations of mixed up train features.

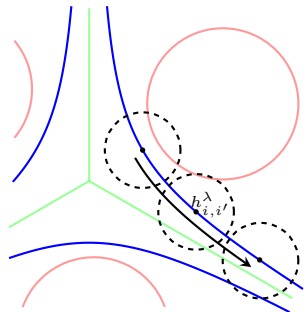

Figure 4: **(Diagram showing the relationship between calibration and the configuration).** As $\lambda$ approaches 0.5, the last-layer activation $h_{i,i'}^\lambda$ (black) traverses the blue line of the configuration, leading to less confident predictions. Simultaneously, the variability of the activation (perforated black circle) results in an increase in misclassification due to the probability of being on the incorrect side of the decision boundary (green) increasing.

## 3 UNCONSTRAINED FEATURES MODEL FOR MIXUP

### 3.1 THEORETICAL CHARACTERIZATION OF OPTIMAL LAST-LAYER FEATURES

To study the resulting last-layer features under mixup, we consider an adaptation of the *unconstrained features model* to mixup training. Let $d \geq C - 1$ be the dimension of the last-layer features, $\boldsymbol{y}_i \in \mathbb{R}^C$ be the one-hot vector in entry $i$, $\boldsymbol{W} \in \mathbb{R}^{C \times d}$ be the classifier, and $\boldsymbol{h}_{ii'}^\lambda \in \mathbb{R}^d$ be the last-layer feature associated with target $\lambda \boldsymbol{y}_i + (1 - \lambda)\boldsymbol{y}_{i'}$. Then, adapting Equation 1 to the unconstrained features setting, we consider the optimization problem given by

$$\min_{\boldsymbol{W}, \boldsymbol{h}_{ii'}^\lambda} \mathbb{E}_{\lambda \sim D_\lambda} \frac{1}{C^2} \sum_{i=1}^{C} \sum_{i'=1}^{C} \left( \mathrm{CE}\left(\boldsymbol{W}\boldsymbol{h}_{ii'}^\lambda, \lambda \boldsymbol{y}_i + (1 - \lambda)\boldsymbol{y}_{i'}\right) + \frac{\lambda_{\boldsymbol{H}}}{2}\|\boldsymbol{h}_{ii'}^\lambda\|_2^2 \right) + \frac{\lambda_{\boldsymbol{W}}}{2}\|\boldsymbol{W}\|_F^2, \quad (2)$$

where $\lambda_{\boldsymbol{W}}, \lambda_{\boldsymbol{H}} > 0$ are the weight decay parameters. It is reasonable to consider decay on the features $\boldsymbol{h}_{ii'}^\lambda$, a practice that is frequently observed in prior work (Zhu et al., 2021; Zhou et al., 2022), due to the implicit decay to the last-layer features from the inclusion of decay in the previous layers' parameters.

The following theorem characterizes the optimal last-layer features under the assumption that the optimal classifier $\boldsymbol{W}$ is a simplex ETF, ie.

$$m\sqrt{\frac{C}{C - 1}}(\boldsymbol{I}_C - \frac{1}{C}\mathbf{1}_C\mathbf{1}_C^\top)\boldsymbol{U}^\top, \quad (3)$$

where $\boldsymbol{I}_C \in \mathbb{R}^{C \times C}$ is the identity, $\mathbf{1}_C \in \mathbb{R}^C$ is the ones vector, $\mathbf{U} \in \mathbb{R}^{d \times C}$ is a partial orthogonal matrix (satisfying $\mathbf{U}^\top \mathbf{U} = \boldsymbol{I}_C$), and $m \in \mathbb{R} \setminus \{0\}$ is its multiplier.

Note that we make this assumption as it holds in practice based on our empirical measurements, illustrated in Figure 5.

**Theorem 3.1.** *Assume that at optimality, $\boldsymbol{W}$ is a simplex ETF with multiplier $m$, and denote the $i$-th row of $\boldsymbol{W}$ by $\boldsymbol{w}_i$. Then, any minimizer of equation 2 satisfies:*

*1) **Same-Class:** For all $i = 1, \ldots, C$ and $\lambda \in [0, 1]$,*

$$\boldsymbol{h}_{ii}^\lambda = \frac{(1 - C)K}{m^2}\boldsymbol{w}_i,$$

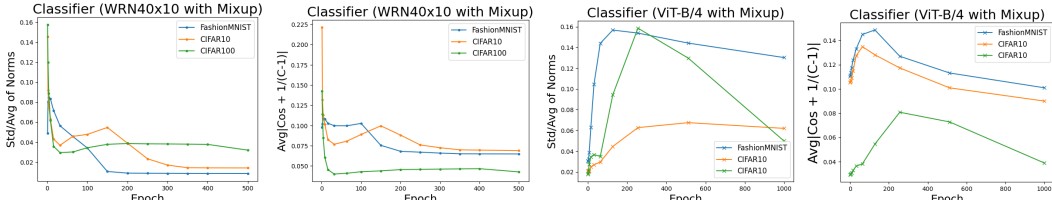

Figure 5: **(Convergence of classifier to simplex ETF)**. Measurements on the classifier, $\boldsymbol{W}$, for each network architecture and dataset combination. **First and third plot:** Coefficient of variation of the classifier norms, $\mathrm{Std}_i\left(\|\boldsymbol{w}_i\|_2\right)/\mathrm{Avg}_i\left(\|\boldsymbol{w}_i\|_2\right)$. **Second and fourth plot:** Standard deviation of the cosines between classifiers of distinct classes, $\mathrm{Std}_{i,i'\neq i}\left(\langle\boldsymbol{w}_i,\boldsymbol{w}_{i'}\rangle/\left(\|\boldsymbol{w}_i\|_2\|\boldsymbol{w}_{i'}\|_2\right)\right)$ with $i\neq i'$. As training progresses, measurements indicate that $\boldsymbol{W}$ is trending toward a simplex ETF configuration.

*where $K < 0$ is the unique solution to the equation*

$$e^{-CK} - \frac{Cm^2}{(1-C)\lambda_{\boldsymbol{H}}K} + C - 1 = 0.$$

*2) **Different-Class:** For all $i\neq i'$ and $\lambda\in[0,1]$,*

$$\boldsymbol{h}_{ii'}^{\lambda} = \frac{(1-C)}{Cm^2}\left(\left(K_\lambda - \langle\boldsymbol{w}_i,\boldsymbol{h}_{ii'}^{\lambda}\rangle\right)\boldsymbol{w}_i + \left((C-1)K_\lambda + \langle\boldsymbol{w}_i,\boldsymbol{h}_{ii'}^{\lambda}\rangle\right)\boldsymbol{w}_{i'}\right),$$

*where $\langle\boldsymbol{w}_i,\boldsymbol{h}_{ii'}^{\lambda}\rangle$ is of the form*

$$\log\left(e^{K_\lambda}\left(2 - C + \frac{Cm^2}{(1-C)K_\lambda\lambda_{\boldsymbol{H}}} \pm \frac{1}{2}\sqrt{\left(C - 2 - \frac{Cm^2}{(1-C)K_\lambda\lambda_{\boldsymbol{H}}}\right)^2 - 4e^{-CK_\lambda}}\right)\right),$$

*and $K_\lambda < 0$ satisfies*

$$e^{\langle\boldsymbol{w}_i,\boldsymbol{h}_{ii'}^{\lambda}\rangle} = \frac{Cm^2}{(1-C)K_\lambda\lambda_{\boldsymbol{H}}}e^{K_\lambda}\left(\frac{(1-C)\lambda_{\boldsymbol{H}}\langle\boldsymbol{w}_i,\boldsymbol{h}_{ii'}^{\lambda}\rangle}{Cm^2} + \lambda\right).$$

The proof of Theorem 3.1 can be found in Appendix A.1

**Interpretation of Theorem.** Theorem 3.1 establishes that, within the framework of our model's assumptions, the optimal same-class features are independent of $\lambda$ and align with the classifier as a simplex ETF. In contrast, the optimal features for different classes are linear combinations (depending on $\lambda$) of the classifier rows corresponding to the mixed-up targets, governed by the above equations. This is consistent with the observations in Figure 1, where the same-class features consistently cluster at simplex vertices, regardless of the value of $\lambda$, while the different-class features dynamically flow between these vertices as $\lambda$ varies.

In Figure 6, we plot the last-layer features obtained from Theorem 3.1, numerically solving for the values of $K$ and $K_\lambda$ that satisfy their respective equations.

Similar to the empirical results in Figure 1, the density of different-class mixup points decreases as $\lambda$ approaches 0 and 1. However, the theoretically optimal features exhibit channels arranged in a hexagonal pattern, differing from the empirical configuration observed in the FashionMNIST and CIFAR10 datasets. In particular, the empirical representations has a more pronounced elongation of different-class features as the mixup parameter $\lambda$ approaches $0.5$. In attempt to understand these differences, we introduce an amplification of these same features in the directions of the classifier rows not corresponding to the mixed-up targets, with increasing amplifications as $\lambda$ gets closer to 0.5 (details of the amplification function are outlined in Appendix A.2). This results in features that behave more similarly to the empirical outcomes, while achieving a very close (though marginally larger) loss when compared to the true optimal configuration (loss values are indicated below each plot in Figure 6). This demonstrates that the features have some degree of flexibility while remaining in close proximity to the minimum loss.

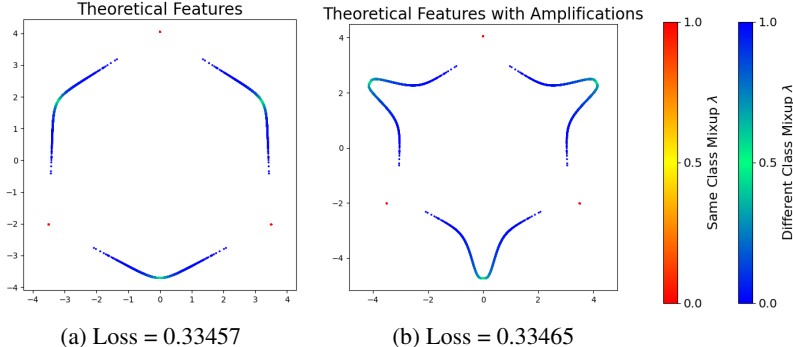

(a) Loss = 0.33457     (b) Loss = 0.33465

Figure 6: **(Optimal activations from unconstrained features model)**. On the left are optimal last-layer activations obtained from our theoretical analysis. We set $m = 3$, $C = 10$, $d = 100$, and $\lambda_{H} = 1 \times 10^{-6}$, and randomly sample 5000 different $\lambda$ values from the $\text{Beta}(1, 1)$ distribution for a randomly selected subset of three classes. Projections are generated using the same method as depicted in Figure 1. Colouring indicates the mixup type (same-class or different-class), and the level of mixup, $\lambda$.

## 3.2 TRAINING WITH FIXED SIMPLEX ETF CLASSIFIER

To further understand the differences between theoretical activations (Figure 6) and empirical activations (Figure 1), we performed an experiment employing mixup within the training framework detailed in Section 2, but fixing the classifier as a simplex ETF. The resulting last-layer features are visualized in Figure 7.

Prior work (Zhu et al., 2021; Yang et al., 2022; Pernici et al., 2022) have explored the effects of fixing the classifier, but not in the context of mixup. Our observations reveal that when the classifier is fixed as a simplex ETF, the empirical features tend to exhibit a more hexagonal shape in its different-class mixup features, aligning more closely with the theoretically optimal features. Moreover, slightly higher generalization performance is achieved when compared to training with a learnable classifier under the same settings.

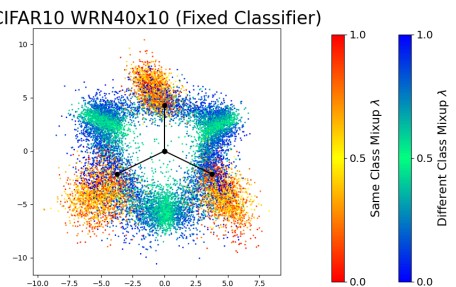

Figure 7: **(Visualization of activations outputted by network trained with mixup fixing the classifier as a simplex ETF)**. Last-layer activations of mixup training data are presented here for a randomly selected subset of three classes. Coloration indicates the type of mixup (same-class or different-class), along with the level of mixup, $\lambda$. This model achieves a test accuracy of 97.35%.

Based on these results, a possible explanation for the variation in configuration is that during training, the classifier is still being learned and requires several epochs to converge to a simplex ETF, as depicted in Figure 5. During this period, the features may traverse regions that lead to slightly suboptimal loss, as there is flexibility in the features' structures without much degradation to the loss performance (depicted in Figure 6).

## 4 RELATED WORK

The success of mixup has prompted many mixup variants, each successful in their own right (Guo et al., 2018; Verma et al., 2019; Yun et al., 2019; Kim et al., 2020). Additionally, various works have been devoted to better understanding the effects and success of the method.

Guo et al. (2018) identified manifold intrusion as a potential limitation of mixup, stemming from discrepancies between the mixed-up label of a mixed-up example and its true label, and they propose a method for overcoming this.

In addition to the work by Thulasidasan et al. (2020) on calibration for networks trained with mixup, Zhang et al. (2022) posits that this improvement in calibration due to mixup is correlated with the capacity of the network. Zhang et al. (2021) theoretically demonstrates that training with mixup corresponds to minimizing an upper bound of the adversarial loss.

Chaudhry et al. (2022) delved into the linearity of various representations of a deep network trained with mixup. They observed that representations nearer to the input and output layer exhibit greater linearity compared to those situated in the middle.

Carratino et al. (2022) interprets mixup as an empirical risk minimization estimator employing transformed data, leading to a process that notably enhances both model accuracy and calibration. Continuing on the same path, Park et al. (2022) offers a unified theoretical analysis that integrates various aspects of mixup methods.

Furthermore, Chidambaram et al. (2021) conducted a detailed examination of the classifier optimal to mixup, comparing it with the classifier obtained through standard training.

Recent work has also been devoted to studying the benefits of mixup with feature-learning based analysis by Chidambaram et al. (2023) and Zou et al. (2023). The former considering two features generated from a symmetric distribution for each class and the latter considering a data model with two features of different frequencies, feature noise, and random noise.

The discovery of Neural Collapse by Papyan et al. (2020) has spurred investigations of this phenomenon. Recent theoretical inquiries by Mixon et al. (2020); Fang et al. (2021); Lu & Steinerberger (2020); E & Wojtowytsch (2020); Poggio & Liao (2020); Zhu et al. (2021); Han et al. (2021); Tirer & Bruna (2022); Wang et al.; Kothapalli et al. (2022) have delved into the analysis of Neural Collapse employing both the unconstrained features model (Mixon et al., 2020) and the layer-peeled model (Fang et al., 2021). Liu et al. (2023) removes the assumption on the feature dimension and the number of classes in Neural Collapse and presents a Generalized Neural Collapse which is characterized by minimizing intra-class variability and maximizing inter-class separability.

To our knowledge, there has not been any investigation into the geometric configuration induced by mixup in the last layer.

## 5 CONCLUSION

In conclusion, through an extensive empirical investigation across various architectures and datasets, we have uncovered a distinctive geometric configuration of last-layer activations induced by mixup. This configuration exhibits intriguing behaviors, such as same-class activations forming a simplex equiangular tight frame (ETF) aligned with their respective classifiers, and different-class activations delineating channels along the decision boundary, with varying densities depending on the mixup coefficient. We also examine the layer-wise trajectory that features follow to reach this configuration in the last-layer, and measure the calibration induced by mixup to provide an explanation for why this particular configuration in beneficial for calibration.

Furthermore, we have complemented our empirical findings with a theoretical analysis, adapting the unconstrained features model to mixup. Theoretical results indicate that the optimal same-class features are independent of the mixup coefficient and align with the classifier, while different-class features are dynamic linear combinations of the classifier rows corresponding to mixed-up targets, influenced by the mixup coefficient. Motivated by our theoretical analysis, we also conduct experiments investigating the configuration of the last-layer activations from training with mixup while keeping the classifier fixed as a simplex ETF. We observe that they align more closely with the theoretically optimal features, with slight improvement in test-performance.

These findings collectively shed light on the intricate workings of mixup in training deep networks, emphasizing its role in organizing last-layer activations for improved calibration. Understanding these geometric configurations induced by mixup opens up avenues for further research into the design of data augmentation strategies and their impact on neural network training.

## 6 ACKNOWLEDGEMENTS

We acknowledge the support of the Natural Sciences and Engineering Research Council of Canada (NSERC). This research was enabled in part by support provided by Compute Ontario (http://www.computeontario.ca/) and Compute Canada (http://www.computecanada.ca/).

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

## A  THEORETICAL MODEL

### A.1  PROOF OF THEOREM 3.1

Our proof uses similar techniques as Yang et al. (2022), but we extend these ideas to the more intricate last-layer features that arise from mixup.

*Proof.* Assuming that $\boldsymbol{W}$ is a simplex ETF with multiplier $m$, our unconstrained features optimization problem in equation 2 becomes separable across $\lambda$ and $i, i'$, and so it suffices to minimize

$$L_{ii'}^{\lambda} = -\lambda \boldsymbol{y}_i \log\left(\frac{e^{\langle \boldsymbol{w}_i, \boldsymbol{h}_{ii'}^{\lambda}\rangle}}{\sum_{k=1}^{C} e^{\langle \boldsymbol{w}_k, \boldsymbol{h}_{ii'}^{\lambda}\rangle}}\right) - (1-\lambda)\boldsymbol{y}_{i'} \log\left(\frac{e^{\langle \boldsymbol{w}_{i'}, \boldsymbol{h}_{ii'}^{\lambda}\rangle}}{\sum_{k=1}^{C} e^{\langle \boldsymbol{w}_k, \boldsymbol{h}_{ii'}^{\lambda}\rangle}}\right) + \frac{1}{2}\lambda_{\boldsymbol{H}} \left\|\boldsymbol{h}_{ii'}^{\lambda}\right\|_2^2$$

over each $\boldsymbol{h}_{ii'}^{\lambda}$ individually.

$$\frac{\partial L_{ii'}^{\lambda}}{\partial \boldsymbol{h}_{ii'}^{\lambda}} = \boldsymbol{W}^{\top}\left(\boldsymbol{p} - (\lambda \boldsymbol{y}_i + (1-\lambda)\boldsymbol{y}_j)\right) + \lambda_{\boldsymbol{H}}\boldsymbol{h}_{ii'}^{\lambda} \qquad (\boldsymbol{p} = \text{softmax}\left(\boldsymbol{W}\boldsymbol{h}_{ii'}^{\lambda}\right))$$

$$= \sum_{j=1}^{C} \boldsymbol{w}_j p_j - \lambda \boldsymbol{w}_i - (1-\lambda)\boldsymbol{w}_{i'} + \lambda_{\boldsymbol{H}}\boldsymbol{h}_{ii'}^{\lambda} \qquad (p_j \text{ j-th entry of } \boldsymbol{p})$$

$$= \sum_{j\neq i,i'} \boldsymbol{w}_j p_j + (p_i - \lambda)\boldsymbol{w}_i + (p_{i'} - (1-\lambda))\boldsymbol{w}_{i'} + \lambda_{\boldsymbol{H}}\boldsymbol{h}_{ii'}^{\lambda}.$$

Setting $\frac{\partial L_{ii'}^{\lambda}}{\partial \boldsymbol{h}_{ii'}^{\lambda}} = 0$ gives

$$\sum_{j\neq i,i'} \boldsymbol{w}_j p_j + (p_i - \lambda)\boldsymbol{w}_i + (p_{i'} - (1-\lambda))\boldsymbol{w}_{i'} + \lambda_{\boldsymbol{H}}\boldsymbol{h}_{ii'}^{\lambda} = 0. \qquad (4)$$

**Case:** $i = i'$

In this case, equation 4 reduces to

$$\sum_{j\neq i} \boldsymbol{w}_j p_j + (p_i - 1)\boldsymbol{w}_i + \lambda_{\boldsymbol{H}}\boldsymbol{h}_{ii}^{\lambda} = 0. \qquad (5)$$

Taking inner product with $\boldsymbol{w}_j, j \neq i$ in equation 5 gives

$$0 = m^2 p_j - \frac{m^2}{C-1}\sum_{k\neq i,j} p_k - \frac{m^2(p_i - 1)}{C-1} + \lambda_{\boldsymbol{H}}\langle \boldsymbol{w}_j, \boldsymbol{h}_{ii}^{\lambda}\rangle$$

$$= m^2 p_j - \frac{m^2}{C-1}\left(\sum_{k\neq j} p_k - 1\right) + \lambda_{\boldsymbol{H}}\langle \boldsymbol{w}_j, \boldsymbol{h}_{ii}^{\lambda}\rangle$$

$$= m^2 p_j \left(\frac{C}{C-1}\right) + \lambda_{\boldsymbol{H}}\langle \boldsymbol{w}_j, \boldsymbol{h}_{ii}^{\lambda}\rangle.$$

Since $m^2, p_j, \frac{C}{C-1}, \lambda_{\boldsymbol{H}}$ are all positive, it follows that $\langle \boldsymbol{w}_j, \boldsymbol{h}_{ii}^{\lambda}\rangle < 0$.

For all $j, k \neq i$, we have

$$\frac{\exp\left(\langle \boldsymbol{w}_j, \boldsymbol{h}_{ii}^{\lambda}\rangle\right)}{\exp\left(\langle \boldsymbol{w}_k, \boldsymbol{h}_{ii}^{\lambda}\rangle\right)} = \frac{p_j}{p_k} = \frac{\langle \boldsymbol{w}_j, \boldsymbol{h}_{ii}^{\lambda}\rangle}{\langle \boldsymbol{w}_k, \boldsymbol{h}_{ii}^{\lambda}\rangle}.$$

Since $\frac{\exp(x)}{x}$ is strictly decreasing on $(-\infty, 0)$, in particular it is injective on $(-\infty, 0)$, so

$$\langle \boldsymbol{w}_j, \boldsymbol{h}_{ii}^\lambda \rangle = \langle \boldsymbol{w}_k, \boldsymbol{h}_{ii}^\lambda \rangle = K,$$

for some $K < 0$ and $p_j = p_k = p$ for all $j, k \neq i$, where

$$p = \lambda_{\boldsymbol{H}} \frac{1 - C}{C} \frac{K}{m^2}.$$

Let $S = \sum_{k=1}^C \exp \langle \boldsymbol{w}_k, \boldsymbol{h}_{ii}^\lambda \rangle$. Then, $\frac{e^K}{S} = p$ and so

$$S = \frac{e^K}{p} = \frac{C}{1 - C} \cdot \frac{m^2}{K\lambda_{\boldsymbol{H}}} e^K.$$

Then,

$$
\begin{aligned}
\boldsymbol{h}_{ii}^\lambda &= \frac{-1}{\lambda_{\boldsymbol{H}}} \left( \sum_{j \neq i} \boldsymbol{w}_j p_j + (p_i - 1) \boldsymbol{w}_i \right) \\
&= \frac{-1}{\lambda_{\boldsymbol{H}}} \left( -p\boldsymbol{w}_i + (p_i - 1) \boldsymbol{w}_i \right) \\
&= \frac{-1}{\lambda_{\boldsymbol{H}}} \left( -p\boldsymbol{w}_i - (C - 1) p\boldsymbol{w}_i \right) \\
&= \frac{1}{\lambda_{\boldsymbol{H}}} Cp\boldsymbol{w}_i \\
&= \frac{(1 - C) K}{m^2} \boldsymbol{w}_i.
\end{aligned}
$$

Taking inner product with $\boldsymbol{w}_i$ in equation 5 gives

$$
\begin{aligned}
0 &= -\frac{m^2}{C - 1} \sum_{k \neq i} p_k + m^2 (p_i - 1) + \lambda_{\boldsymbol{H}} \langle \boldsymbol{w}_i, \boldsymbol{h}_{ii}^\lambda \rangle \\
&= m^2 p_i \left( \frac{1}{C - 1} + 1 \right) - m^2 \left( \frac{1}{C - 1} + 1 \right) + \lambda_{\boldsymbol{H}} \langle \boldsymbol{w}_i, \boldsymbol{h}_{ii}^\lambda \rangle \\
&= m^2 \frac{C}{C - 1} (p_i - 1) + \lambda_{\boldsymbol{H}} \langle \boldsymbol{w}_i, \boldsymbol{h}_{ii}^\lambda \rangle \\
&= m^2 \frac{C}{C - 1} (-(C - 1) p) + \lambda_{\boldsymbol{H}} \langle \boldsymbol{w}_i, \boldsymbol{h}_{ii}^\lambda \rangle,
\end{aligned}
$$

and so

$$\langle \boldsymbol{w}_i, \boldsymbol{h}_{ii}^\lambda \rangle = (1 - C)K. \tag{6}$$

By our definition of $\boldsymbol{p}$ as the softmax applied on $\boldsymbol{W} \boldsymbol{h}_{ii}^\lambda$, we have

$$
\begin{aligned}
e^{\langle \boldsymbol{w}_i, \boldsymbol{h}_{ii'}^\lambda \rangle} &= Sp_i \\
&= \frac{C}{1 - C} \cdot \frac{m^2}{K\lambda_{\boldsymbol{H}}} e^K (1 - (C - 1)p) \\
&= \frac{C}{1 - C} \cdot \frac{m^2}{K\lambda_{\boldsymbol{H}}} e^K \left( 1 + \frac{((C - 1)^2 \lambda_{\boldsymbol{H}} K}{Cm^2} \right).
\end{aligned}
$$

ie.

$$e^{-CK} = \frac{Cm^2}{(1 - C)\lambda_{\boldsymbol{H}} K} + 1 - C.$$

So, $K$ must satisfy $f(K) = 0$, where $f \colon (-\infty, 0) \to \mathbb{R}$ is defined by

$$f(x) = e^{-Cx} - \frac{Cm^2}{(1 - C)\lambda_{\boldsymbol{H}} x} - (1 - C),$$

(note that we only consider the domain $(-\infty, 0)$ since we've shown that $K < 0$). We will show that there exists a unique $K$ satisfying these properties.

$$f'(x) = -Ce^{-Cx} + \frac{Cm^2}{(1-C)\lambda_H x^2} < 0,$$

since $-Ce^{-Cx} < 0$ and $\frac{Cm^2}{(1-C)\lambda_H x^2} < 0$ (since all terms in the product are positive except $(1-C) < 0$) for all $x$. So $f$ is strictly decreasing, and thus it is injective.

$\lim_{x \to 0^-} f(x) = -\infty$ and $\lim_{x \to -\infty} f(x) = \infty$, so by continuity of $f$, there exists $K < 0$ such that $f(K) = 0$, and $K$ is unique by injectivity of $f$.

**Case:** $i \neq i'$

Taking inner product with $w_j, j \neq i, i'$ in equation 4 and using properties of $W$ as a simplex ETF gives

$$
\begin{aligned}
0 &= m^2 p_j - \frac{m^2}{C-1} \sum_{k \neq i, i', j} p_k - \frac{m^2 (p_i - \lambda)}{C-1} - \frac{m^2 (p_{i'} - (1-\lambda))}{C-1} + \lambda_H \left\langle w_j, h_{ii'}^\lambda \right\rangle \\
&= m^2 p_j - \frac{m^2}{C-1} \left( \sum_{k \neq j} p_k - 1 \right) + \lambda_H \left\langle w_j, h_{ii'}^\lambda \right\rangle \\
&= m^2 p_j \left( 1 + \frac{1}{C-1} \right) + \lambda_H \left\langle w_j, h_{ii'}^\lambda \right\rangle \\
&= m^2 p_j \left( \frac{C}{C-1} \right) + \lambda_H \left\langle w_j, h_{ii'}^\lambda \right\rangle.
\end{aligned}
$$

By the same argument as in the previous case, we get that for all $j, k \neq i, i'$,

$$\left\langle w_j, h_{ii'}^\lambda \right\rangle = \left\langle w_k, h_{ii'}^\lambda \right\rangle = K_\lambda,$$

for some $K_\lambda < 0$ (we will omit the subscript $\lambda$ for brevity as we are optimizing over each $\lambda$ individually). Thus $p_j = p_k = p$ for all $j, k \neq i, i'$, where

$$p = \lambda_H \frac{1-C}{C} \frac{K}{m^2}.$$

Let $S = \sum_{k=1}^C \exp \left\langle w_k, h_{ii'}^\lambda \right\rangle$. Then, $\frac{e^K}{S} = p$ and so

$$S = \frac{e^K}{p} = \frac{C}{1-C} \cdot \frac{m^2}{K\lambda_H} e^K. \tag{7}$$

Taking inner product with $w_i$ in equation 4 gives

$$
\begin{aligned}
0 &= \frac{-m^2}{C-1} \sum_{j \neq i, i'} p_j + m^2 (p_i - \lambda) - \frac{m^2}{C-1} (p_{i'} - (1-\lambda)) + \lambda_H \left\langle w_i, h_{ii'}^\lambda \right\rangle \\
&= \frac{-m^2}{C-1} (1 - p_i - p_{i'}) + m^2 (p_i - \lambda) - \frac{m^2}{C-1} (p_{i'} - (1-\lambda)) + \lambda_H \left\langle \omega_i, h_{ii'}^\lambda \right\rangle \\
&= \frac{m^2}{C-1} p_i + m^2 (p_i - \lambda) - \frac{m^2}{C-1} \lambda + \lambda_H \left\langle w_i, h_{ii'}^\lambda \right\rangle \\
&= m^2 p_i \left( 1 + \frac{1}{C-1} \right) - m^2 \lambda \left( 1 + \frac{1}{C-1} \right) + \lambda_H \left\langle w_i, h_{ii'}^\lambda \right\rangle \\
&= m^2 \left( 1 + \frac{1}{C-1} \right) (p_i - \lambda) + \lambda_H \left\langle w_i, h_{ii'}^\lambda \right\rangle.
\end{aligned}
$$

So,

$$\frac{Cm^2}{C-1} (p_i - \lambda) + \lambda_H \left\langle \omega_i, h_{ii'}^\lambda \right\rangle = 0. \tag{8}$$

Similarly, taking inner product with $\boldsymbol{w}_{i'}$ in equation 4 gives us

$$
\begin{aligned}
0 &= \frac{-m^2}{C-1} \sum_{j \neq i, i'} p_j - \frac{m^2}{C-1} (p_i - \lambda) + m^2 (p_{i'} - (1-\lambda)) + \lambda_{\boldsymbol{H}} \langle \boldsymbol{w}_{i'}, \boldsymbol{h}_{ii'}^{\lambda} \rangle \\
&= m^2 \left(1 + \frac{1}{C-1}\right) (p_{i'} - (1-\lambda)) + \lambda_{\boldsymbol{H}} \langle \boldsymbol{w}_{i'}, \boldsymbol{h}_{ii'}^{\lambda} \rangle,
\end{aligned}
$$

and so

$$
\frac{Cm^2}{C-1} (p_{i'} - (1-\lambda)) + \lambda_{\boldsymbol{H}} \langle \boldsymbol{w}_{i'}, \boldsymbol{h}_{ii'}^{\lambda} \rangle = 0. \tag{9}
$$

Summing equation 8 and equation 9 gives

$$
\begin{aligned}
0 &= \frac{Cm^2}{C-1} (p_i + p_{i'} - 1) + \lambda_{\boldsymbol{H}} \left( \langle \boldsymbol{w}_i, \boldsymbol{h}_{ii'}^{\lambda} \rangle + \langle \boldsymbol{w}_{i'}, \boldsymbol{h}_{ii'}^{\lambda} \rangle \right) \\
&= \frac{Cm^2}{C-1} (-(C-2)p) + \lambda_{\boldsymbol{H}} \left( \langle \boldsymbol{w}_i, \boldsymbol{h}_{ii'}^{\lambda} \rangle + \langle \boldsymbol{w}_{i'}, \boldsymbol{h}_{ii'}^{\lambda} \rangle \right) \\
&= (C-2)\lambda_{\boldsymbol{H}} K + \lambda_{\boldsymbol{H}} \left( \langle \boldsymbol{w}_i, \boldsymbol{h}_{ii'}^{\lambda} \rangle + \langle \boldsymbol{w}_{i'}, \boldsymbol{h}_{ii'}^{\lambda} \rangle \right),
\end{aligned}
$$

and so

$$
\langle \boldsymbol{w}_i, \boldsymbol{h}_{ii'}^{\lambda} \rangle + \langle \boldsymbol{w}_{i'}, \boldsymbol{h}_{ii'}^{\lambda} \rangle = -(C-2)K.
$$

Then, using equation 7 and the definition of $S$ gives us

$$
\begin{aligned}
\frac{C}{1-C} \cdot \frac{m^2}{K\lambda_{\boldsymbol{H}}} e^K = S &= \sum_{k=1}^{C} e^{\langle \boldsymbol{w}_k, \boldsymbol{h}_{ii'}^{\lambda} \rangle} \\
&= (C-2)e^K + e^{\langle \boldsymbol{w}_i, \boldsymbol{h}_{ii'}^{\lambda} \rangle} + e^{-(C-2)K - \langle \boldsymbol{w}_i, \boldsymbol{h}_{ii'}^{\lambda} \rangle},
\end{aligned}
$$

and thus

$$
\left( e^{\langle \boldsymbol{w}_i, \boldsymbol{h}_{ii'}^{\lambda} \rangle} \right)^2 + e^K \left( C - 2 - \frac{C}{1-C} \cdot \frac{m^2}{K\lambda_{\boldsymbol{H}}} \right) e^{\langle \boldsymbol{w}_i, \boldsymbol{h}_{ii'}^{\lambda} \rangle} + e^{-(C-2)K} = 0. \tag{10}
$$

We then solve the quadratic equation in $e^{\langle \boldsymbol{w}_i, \boldsymbol{h}_{ii'}^{\lambda} \rangle}$ to get

$$
e^{\langle \boldsymbol{w}_i, \boldsymbol{h}_{ii'}^{\lambda} \rangle} = \frac{-e^K \left( C - 2 - \frac{Cm^2}{(1-C)K\lambda_{\boldsymbol{H}}} \right) \pm \sqrt{\left( e^K \left( C - 2 - \frac{Cm^2}{(1-C)K\lambda_{\boldsymbol{H}}} \right) \right)^2 - 4e^{-(C-2)K}}}{2}.
$$

So, $\langle \boldsymbol{w}_i, \boldsymbol{h}_{ii'}^{\lambda} \rangle$ is of the form

$$
\log \left( \frac{-e^K \left( C - 2 - \frac{Cm^2}{(1-C)K\lambda_{\boldsymbol{H}}} \right) \pm \sqrt{\left( e^K \left( C - 2 - \frac{Cm^2}{(1-C)K\lambda_{\boldsymbol{H}}} \right) \right)^2 - 4e^{-(C-2)K}}}{2} \right).
$$

By our definition of $\boldsymbol{p}$ as the softmax applied on $\boldsymbol{W} \boldsymbol{h}_{ii'}^{\lambda}$, $K$ must satisfy

$$
\begin{aligned}
e^{\langle \boldsymbol{w}_i, \boldsymbol{h}_{ii'}^{\lambda} \rangle} &= Sp_i \\
&= \frac{e^K}{p} \\
&= \frac{C}{1-C} \cdot \frac{m^2}{K\lambda_{\boldsymbol{H}}} e^K \left( \frac{(1-C)\lambda_{\boldsymbol{H}} \langle \boldsymbol{w}_i, \boldsymbol{h}_{ii'}^{\lambda} \rangle}{Cm^2} + \lambda \right).
\end{aligned}
$$

We have

$$
\sum_{j \neq i, i'} \boldsymbol{w}_i p_j + (p_i - \lambda) \, \boldsymbol{w}_i + (p_{i'} - (1 - \lambda)) \, \boldsymbol{w}_{i'}
$$

$$
= p \left( -\boldsymbol{w}_i - \boldsymbol{w}_{i'} \right) + (p_i - \lambda) \, \boldsymbol{w}_i + (p_{i'} - (1 - \lambda)) \, \boldsymbol{w}_{i'} \qquad \text{(since } \sum_{j=1}^C \boldsymbol{w}_j = 0\text{)}
$$

$$
= (p_i - \lambda - p) \, \boldsymbol{w}_i + (p_{i'} - (1 - \lambda) - p) \, \boldsymbol{w}_{i'}.
$$

Substituting this into equation 4, we get

$$
\boldsymbol{h}_{ii'}^{\lambda} = \frac{-1}{\lambda_{\boldsymbol{H}}} \left( (p_i - \lambda - p) \, \boldsymbol{w}_i + (p_{i'} - (1 - \lambda) - p) \, \boldsymbol{w}_{i'} \right)
$$

$$
= \frac{1}{\lambda_{\boldsymbol{H}}} \left( (p - (p_i - \lambda)) \, \boldsymbol{w}_i + (p - (p_{i'} - (1 - \lambda))) \, \boldsymbol{w}_{i'} \right)
$$

$$
= \frac{(1-C)K\lambda_{\boldsymbol{H}} - (1-C)\lambda_{\boldsymbol{H}} \left\langle \boldsymbol{w}_i, \boldsymbol{h}_{ii'}^{\lambda} \right\rangle}{\lambda_{\boldsymbol{H}} C m^2} \boldsymbol{w}_i + \frac{(1-C)K\lambda_{\boldsymbol{H}} - (1-C)\lambda_{\boldsymbol{H}} \left\langle \boldsymbol{w}_{i'}, \boldsymbol{h}_{ii'}^{\lambda} \right\rangle}{\lambda_{\boldsymbol{H}} C m^2} \boldsymbol{w}_{i'}
$$

$$
= \frac{(1-C)}{C m^2} \left( \left( K - \left\langle \boldsymbol{w}_i, \boldsymbol{h}_{ii'}^{\lambda} \right\rangle \right) \boldsymbol{w}_i + \left( K - \left\langle \boldsymbol{w}_{i'}, \boldsymbol{h}_{ii'}^{\lambda} \right\rangle \right) \boldsymbol{w}_{i'} \right)
$$

$$
= \frac{(1-C)}{C m^2} \left( \left( K - \left\langle \boldsymbol{w}_i, \boldsymbol{h}_{ii'}^{\lambda} \right\rangle \right) \boldsymbol{w}_i + \left( (C-1)K + \left\langle \boldsymbol{w}_i, \boldsymbol{h}_{ii'}^{\lambda} \right\rangle \right) \boldsymbol{w}_{i'} \right),
$$

$\square$

### A.2 Amplification of Theoretical features

In this section we provide additional details of function used to generate the amplified features in Figure 6.

We define $\epsilon(\lambda) = \frac{4}{5} \exp(-20(\lambda - 0.5)^4) - \frac{2}{5}$. Then for the different class features ($i \neq i'$), define the amplified features as $\tilde{\boldsymbol{h}}_{ii'}^{\lambda} = \boldsymbol{h}_{ii'}^{\lambda} - \epsilon(\lambda) \sum_{j \neq i, i'} \boldsymbol{w}_j$.

The motivation for the function form of $\epsilon(\lambda)$ is to ensure that it is symmetric about $\lambda = 0.5$, increasing when $\lambda < 0.5$ and decreasing when $\lambda > 0.5$ with its maximum at $\lambda = 0.5$. These properties correspond to a larger amplification as $\lambda$ approaches $0.5$, while preserving symmetry in the amplifications. Note that the exact function $\epsilon(\lambda)$ is not important, but rather that it yields last-layer features that are closer to the empirical results (with elongations), while resulting in just a minor increase in loss. As mentioned in the main text, this implies that the features can have some deviation from the theoretical optimum without much change in the value of the loss.

## B Experimental details

### B.1 Hyperparameter settings

For the WideResNet experiments, we minimize the mixup loss using stochastic gradient descent (SGD) with momentum $0.9$ and weight decay $1 \times 10^{-4}$. All datasets are trained on a WideResNet-40-10 for 500 epochs with a batch size of 128. We sweep over 10 logarithmically spaced learning rates between $0.01$ and $0.25$, picking whichever results in the highest test accuracy. The learning rate is annealed by a factor of 10 at 30%, 50%, and 90% of the total training time.

For the ViT experiments, we minimize the mixup loss using Adam optimization (Kingma & Ba, 2017). For each dataset we train a ViT-B with a patch size of 4 for 1000 epochs with a batch size of 128. We sweep over 10 logarithmically spaced learning rates from $1 \times 10^{-4}$ to $3 \times 10^{-3}$ and weight decay values from $0$ to $0.05$, selecting whichever yields the highest test accuracy. The learning rate is warmed up for 10 epochs and is annealed using cosine annealing as a function of total epochs.

### B.2 Projection Method

For all of the last-layer activation plots, the same projection method is used. First, we randomly select three classes. We denote the centred last-layer activations for said classes by a matrix $H \in$

$\mathbb{R}^{m \times n}$ and the classifier of the network for said classes as $W \in \mathbb{R}^{3 \times m}$. The projection method is then as follows:

1. Calculate $USV^T = \text{SVD}(W^*)$ where $W^*$ is the normalized classifier.
2. Define $Q = UV^T$
3. Let $A \in \mathbb{R}^{2 \times 3}$ be a two dimensional representation of a three dimensional simplex.
4. Compute $X = AQH$ and plot.

## C    EXPECTED CALIBRATION ERROR

To calculate the expected calibration error, first gather the predictions into $M$ bins of equal interval size. Let $B_m$ be the set of predictions whose confidence is in bin $m$. We can define the accuracy and confidence of a given bin as

$$\text{acc}(B_m) = \frac{1}{|B_m|} \sum_{i \in B_m} \mathbf{1}\left(\hat{y}_i = y_i\right)$$

$$\text{conf}(B_m) = \frac{1}{|B_m|} \sum_{i \in B_m} \hat{p}_i$$

where $\hat{p}_i$ is the confidence of example $i$. The expected calibration (ECE) is then calculated as

$$\text{ECE} = \sum_{m=1}^{M} \frac{|B_m|}{n} \left|\text{acc}\left(B_m\right) - \text{conf}\left(B_m\right)\right|$$

## D    ADDITIONAL LAST-LAYER PLOTS

Here we provide additional plots of last-layer activations. Namely, Figure 8 provides additional baseline last-layer activation plots for mixup data for every architecture and dataset combination in Figure 1. Figure 9 provides last-layer activations for the additional $\alpha$ value in Table 2. Figure 10 shows the evolution of the last-layer activations throughout training. Figure 11 shows last-layer activations for multiple random subsets of three classes. Finally, Figure 12 shows the last-layer activations for a ViT-B/4 trained on CIFAR10 using the AdamW optimizer.

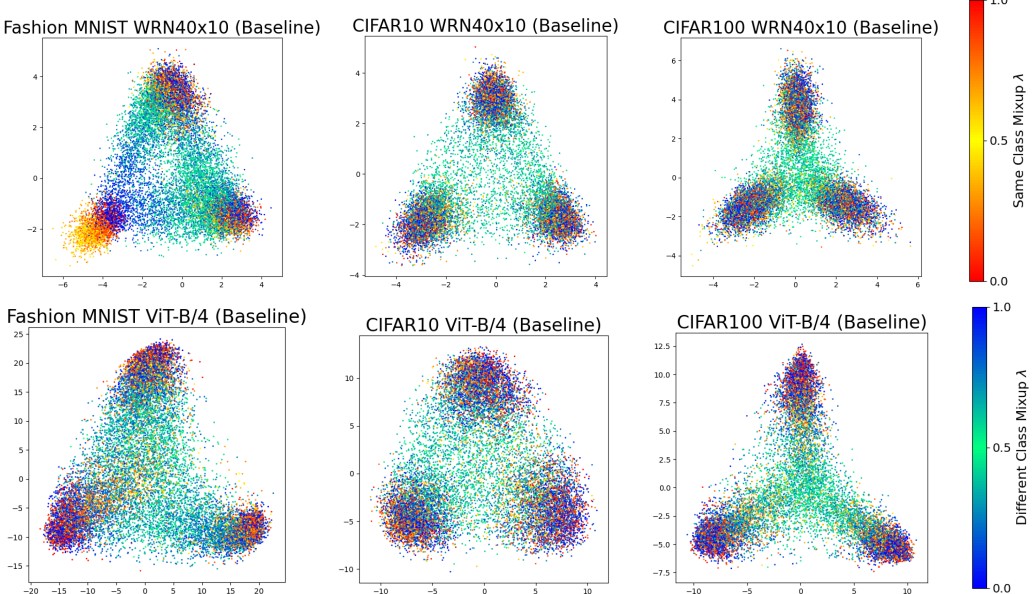

Figure 8: **(Visualization of activations outputted by networks trained without mixup)**. Last-layer activations for a randomly selected subset of three classes of mixup training data for various dataset and network architecture combinations. Projections are generated using the same method as Figure 1. All networks are trained using empirical risk minimization (no mixup). Colouring indicates mixup type (same-class or different-class), and the level of mixup, $\lambda$.

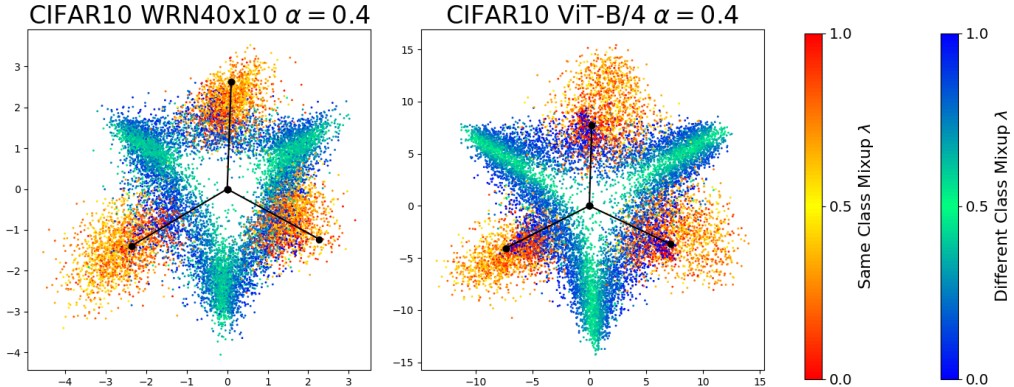

Figure 9: **(Visualization of activations outputted by networks trained with $\alpha = 0.4$)**. Last-layer activations for WideResNet-40-10 and ViT-B/4 trained on the CIFAR10 dataset, subsetted to three randomly selected classes. Projections are generated using the same method as Figure 1. For both cases, $\alpha = 0.4$ is used. Colouring indicates mixup type (same-class or different-class), and the level of mixup, $\lambda$. Relevant classifiers plotted in black.

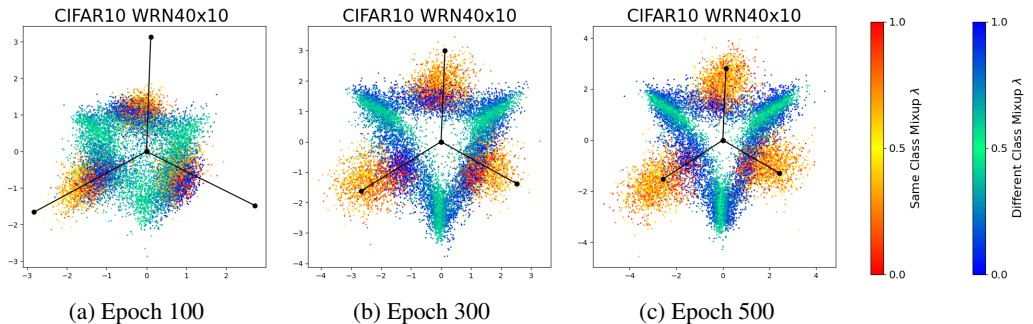

(a) Epoch 100          (b) Epoch 300          (c) Epoch 500

Figure 10: **(Activation convergence during training with mixup)**. Evolution of Last-layer activations for WideResNet-40-10 trained on CIFAR10 through training. Projections are generated in the same manner as Figure 1. Coloration indicates the type of mixup (same-class or different-class), along with the level of mixup, $\lambda$. Relevant classifiers plotted in black. As training progresses, different-class mixup points are pushed towards the decision boundary converging to the configuration depicted in Figure 1.

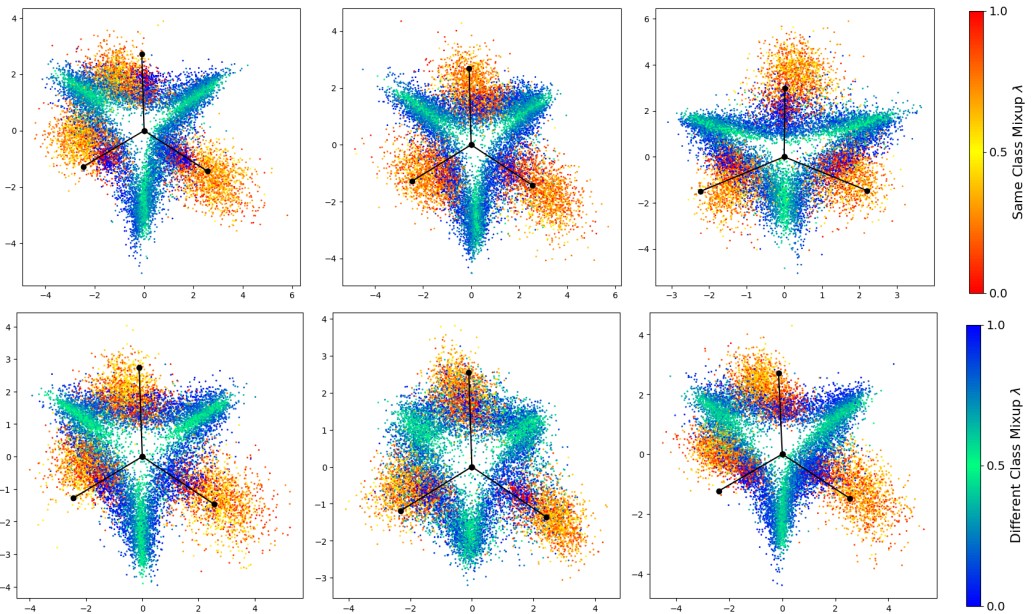

Figure 11: **(Visualization of last-layer activations for multiple subsets of classes)**. Last-layer activations for a randomly selected subsets of three classes of mixup training data for WRN-40-10 trained on CIFAR10. Projections are generated using the same method as Figure 1. Colouring indicates mixup type (same-class or different-class), and the level of mixup, $\lambda$. Black indicates relevant classifiers.

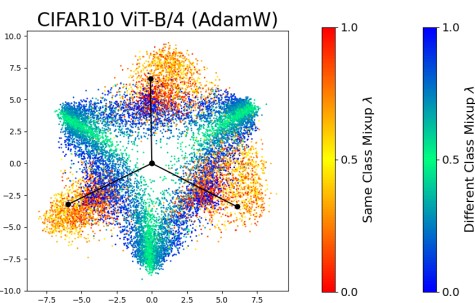

Figure 12: **(Visualization of activations outputted by ViT-B trained with mixup using AdamW)**. Last-layer activations for ViT-B/4 trained following same training regiment as the ViT's outlined in section 2 except with the AdamW optimizer. Projection is generated using the same method as Figure 1. Colouring indicates mixup type (same-class or different-class), and the level of mixup, $\lambda$. Black indicates relevant classifiers.

