# OpenReview forum: "Pushing Boundaries: Mixup's Influence on Neural Collapse"
_ICLR.cc/2024/Conference — ICLR 2024 poster_

### Official Review · Reviewer_nn4q · 2023-10-30

**Soundness:** 3 good
**Presentation:** 3 good
**Contribution:** 3 good
**Rating:** 6
**Confidence:** 4

**Summary:**

The study investigates the phenomenon of Neural Collapse within the mixup training regime. Theoretically, the research employs a modified unconstrained feature model, distinguishing features based on whether they originate from mixup samples of identical or distinct classes and characterize the geometric structure at convergence. The work also conducts experiments consisting of different network architectures and datasets to corroborate their findings.

**Strengths:**

1. The paper is clearly written and well-motivated, the topic covered in the paper is novel to the best of the review's knowledge.
2. The introduction of the modified unconstrained feature model is clear and the theoretical results in the paper are well discussed and easy to understand.

**Weaknesses:**

Despite stating the study aims to explore how the proven geometric configurations in the paper can shed light on the success of mixup, the current iteration of the paper falls short in this regard. Mixup is a well-established augmentation technique with a lot of empirical success (e.g., generalization, adversarial robustness mentioned in this work), but it's hard for the reviewer to establish the connections between these empirical successes and the theoretical results shown in the paper. The authors talk about the difference in test performance of FashionMNIST might be attributed to its inability to achieve the theoretical geometric structure. Yet, such results are not definitive on their own. Therefore, the reviewer believes the paper needs additional work to discuss its practical implications.

Minor:
Sections 1.2-1.5 could be placed before the figures on pages 2 and 3 for better clarity.

**Questions:**

1. Why do the authors keep the models in train mode when plotting? Does this make a difference in the results?

---

> ### Author Response · Authors · 2023-11-18
> **Rebuttal by Authors**
>
> The authors would like to extend their gratitude to the reviewer for their constructive response. Below we respond to the weaknesses and questions.
>
> Weaknesses:
>
> - We have updated the paper to include more experimentation and analysis relating to calibration. We believe that this addition provides a connection between our initial findings and the empirical success of mixup.
> We have updated the formatting of the paper in the way that the reviewer has suggested to improve the clarity.
>
> Questions:
> - Train mode was chosen mostly for consistency among plots. Yes, there is a slight difference in the plots for the resnets. We have included a figure for the network in eval mode in the appendix to show this difference. We believe this difference is due to batch normalization but we have not proved such a claim.
>
>
> We again thank the reviewer for the time and effort spent reading and reviewing our paper. We offer a summary of the overall changes made to the manuscript below.
>
>
> ### 1. **Enhanced Paper Structure and Readability**:
>    - Figures are now strategically placed closer to the relevant text for a coherent reading experience.
>    - Text from figure captions is integrated into the main body for better context.
>    - Expanded the experiments section to provide more comprehensive insights.
>
> ### 2. **Additional Experiments, Analyses, and Figures**:
>    - Updated the colors in Figure 1 for better visualization and added to it the classifiers.
>    - Updated Figure 2, showcasing the feature configuration with MSE loss training, and its comparison with CE loss. This is meant to emphasize how the configuration obtained from the MSE loss is expected and uninteresting whereas the one obtained from CE, which is right next to it, is very different and unexpected.
>    - Introduced Figure 3, illustrating the layer-wise trajectories of the CLS token for a ViT trained with mixup.
>    - Introduced Table 2 reporting network calibration across various scenarios.
>    - Introduced Figure 4 which elucidates the geometric configuration's role in improving calibration.
>    - Introduced Figure 7 which presents changes in the mixup feature configuration once the last-layer classifier is fixed throughout training.
>
> ### 3. **Addressing Specific Reviewer Concerns**:
>    - Added detailed explanation of 2D projection methods used in our visualizations.
>    - Expanded on the theoretical framework, particularly regarding the assumptions about the simplex ETF classifier and its implications.
>    - Investigated the layer-wise trajectory of representations and its implications in mixup training.
>    - Discussed the practical significance of our findings in the context of calibration improvements.
>
> Through these revisions, we aim to bridge the gap between the practical success of mixup training and its theoretical understanding, providing a more comprehensive and valuable contribution to the field. In light of these refinements, should you find it fitting, we would be most grateful for any potential reconsideration of our score.

---

> > ### Comment · Reviewer_nn4q · 2023-11-20
> >
> > Thanks to the authors for the updated paper and the rebuttal. The additional experiments and demonstrations regarding Calibration are interesting and enhance the connection between the theoretical framework and empirical observations, per the reviewer's concern. However,  if I am not mistaken, all the calibration results are newly added in the rebuttal period which makes the paper change quite dramatically from the initial submission. According to the ICLR Author guidelines, which state that 'The revised version shouldn’t read like a different paper compared to your original abstract submission,' it seems the paper may need a fresh round of review given the newly added results.
> >
> > With that being said, the reviewer still appreciates the current updated version and its findings, hence raising the score 5-> 6. Thanks for the response again.

---

### Official Review · Reviewer_vgCZ · 2023-10-30

**Soundness:** 2 fair
**Presentation:** 2 fair
**Contribution:** 1 poor
**Rating:** 6
**Confidence:** 3

**Summary:**

The paper investigated the neural collapse phenomenon in mixup training. Specifically, it conducted experiments to show for synthetic examples mixed from same classes, their last layer activations align with the classifier; for those mixed from different classes, their last layer activations cluster around the decision boundaries.

The paper also explained this phenomenon by studying a unconstrained feature model, showing that under certain assumptions on the classifier, the optimal last layer activations for same-classed mixed examples align with the classifier, while those for ddifferen-class imxedd examples are linear combinations of components of the classifier.

**Strengths:**

1. Conducted investigations of neural collapse in mixup scenario, which is not previously explored
2. Adequate experiments and clear visualizations of the results
3. Theoretical explanation of the observed phenomenons is provided and proved.

**Weaknesses:**

1. Poor layout of the text and figures. The figures don't align well with their appearances in the sections.

2. The connection between neural collapse in mixup and the generalization and calibration improvement of mixup is not further investigated

3. In Figures 1, 2, 3, 5 and 7, although the literal descriptions in the captions are clear, but in the plots the gradient effects of the colors are not distinct. It hard to tell how the $\lambda$'s vary from the plots.

4. Also in these figures, both the same-class and the different-class activations with $\lambda=0.5$ use the color black. Again, though the plots are clearly described in words, the colorations may still bring misunderstandings.

5. How the activations are projected into 2D planes for visualization in the figures is not clearly explained. Particularly:
 a. Is it training data or test data that is used to generated these plots?
 b. In the caption of Figure 1, what does it mean by saying "project activations onto the classifier"? From my understanding of the initial observation of neural collapse, we just need to project the last layer activations and the components of the classifier onto a same low-dimension space. Is that correct? And if so is the projecting method in this paper different from that?
 c. How is the conclusion "last layer activations align with the classifier" justified from the figures? Or from other observations presented in this paper?

6. The theoretical explanation has made an assumption that the classifier is a simplex ETF. This would be a very strong assumption if further proof or demonstration of it is not provided.

7. Also, the theoretical explanation only considers the last layer features and the training targets, but not the inputs. In mixup, sometimes the mixed training target of a mixed input may not in fact be the ground-truth target of the input. Consider a datasets with three clusters of points side by side, suppose the clusters on the two sides are class 1 and the cluster in the middle is class 2. It is possible that a point mixed from two points in class 1 fall into the cluster of class 2, in other words, this point may be labelled both class 1 or class 2. These two scenarios will result in different training target, which will then result in different optimal last layer features. However, the point can have only one possible last layer feature output in a single model. Does this contradict the theoretical explanations in this paper? Should this point be aligned with the classifier component corresponding to class 1 or that corresponding to class 2?

8. In section 4.4, I think the observations in this paper don't sufficiently corroborate the so mentioned linearity of representations, since this paper didn't investigate the representations in early or middle layers. If only based on the observations of this paper, one can also argue that the representations in all layers have linearity in mixup.

9. Overall I think, although the reported observations are new, they are nevertheless superficious and brief. In my opinion it's not much of surprise that same-class  mixed activations perform similarly to conventional neural collapse in ERM and that different-class mixed activations align with the decision boundaries, since different-class mixup induces linear combinations on the training targets while same-class mixup normally somehow performs similarly to conventional data augmentations. The effort the authors have put into the investigations is appreciated, but the presented results don't seem adequate to constitute a rich-content formal paper.

**Questions:**

1. In section 1.5, how do the authors justify if a traineed model shows "good generalization"? Is it based on some well defined metrics or solely based on intuition?

2. Figure 6, second row. The characteristics exhibit an interesting behavior, that they are low at the very beginning of the training processes, then increase, and then decrease again, and their final level may even be higher than their initial level. Can the authors give some explanations or intuitive insights of this trend?

3. Section 4.2. Again, how is the statement "... reaped the minimum benefits" is justified?

4. Section 4.2. "... potential correlation ...". Is there any further investigations into thie potential correlation? Or is there any intuitions reasons of it?

5. Section 4.2. What is channel collapse?

6. How do the observed phenomenons help explain mixup's working mechanism in improving generalization and calibration? In fact, I think the calibration performance is barely mentioned in the main context.

---

> ### Author Response · Authors · 2023-11-18
> **Rebuttal by Authors 1/2**
>
> We would like to express our gratitude to the reviewer for the time and effort put into their assessment of our paper. In the following response, we will address each of the weaknesses and questions raised. First the weaknesses:
>
> 1, 3,  4. Upon your suggestion, we have made significant changes to the formatting of the paper as well as the coloration of the plots presented. We hope that this new format and visualization have resolved any readability issues you encountered.
>
> 2. This is a valid point to raise, and in response to it, we have added an additional section investigating and explaining the relationship that this configuration has with calibration.
>
> 5.  We have added additional descriptions of the last-layer activations being plotted as well as a detailed explanation of the projection method in the appendix. You are correct about the Neural Collapse projections. The only difference between ours is that in the Neural Collapse paper, to the best of our knowledge, everything is projected onto the span of the class means, whereas we are projecting onto the span of the classifiers. Indeed, we have neglected to display how the shape aligns with the classifier so we have updated the figures to include the classifiers.
>
> 6. It would be a strong assumption, so we have incorporated experiments illustrating the convergence of the classifier to a simplex ETF (Figure 5). However, we acknowledge the importance of theoretical proof of the characterization of optimal features without making this assumption, and we have designated this as a direction for future work.
>
> 7. If the scenario given by the reviewer were to occur, then let $x_1$ and $x_1’$ be training examples of class 1, $\tilde{\lambda}$ be such that $\tilde{\lambda}x_1 + (1-\lambda)x_1’ = x$, a training example of class 2.
>
> Our objective function involves an expectation over $\lambda$, and since the set {$\{ \tilde{\lambda} \}$} is of Lebesgue measure 0, the case that $x$ is labeled as class 1 is negligible compared to when $x$ is assigned class 2, so $x$ would be aligned with class 2. Thus, we don’t consider this in our theoretical results.
>
> Since in practice lambda is sampled, we can also consider a more discrete approach. This would result in two terms involving the input $x$ contributing to the overall loss, one with label $y_1$ and the other with label $y_2$. That is, there would be a term $\operatorname{CE}(x, y_1)$ and another term $\operatorname{CE}(x, y_2)$. If the first term appears with probability $P$ and the second with probability $(1-P)$, then overall we get:
>
> $P \operatorname{CE}(x, y_1) + (1-P) \operatorname{CE}(x, y_2) = \operatorname{CE}(x, Py_1 + (1-P)y_2)$
>
> Then, the example overall will have an effective label of $Py_1 + (1-P)y_2$, which corresponds to a mixup label with mixup coefficient P, which is covered by our theoretical result.
>
> Interestingly, [1] considers the general setting proposed by the reviewer called manifold intrusion. There the authors claimed that mixup can be detrimental for cases where mixed up examples align with the original training data, exactly the same way the reviewer is proposing.
> The authors showed that this can occur in datasets like MNIST and that this causes a deterioration in test performance. However, it is unclear to what extent this occurs in more complicated datasets which is why in our paper we did not consider this case.
>
> 8.  To further investigate the layer-wise trajectory of representations, we have provided an additional experiment doing a layer-wise projection of the CLS token for a ViT trained with mixup. We have also modified the discussion section being referenced to more accurately reflect the results of the paper.
>
> 9. We appreciate the different perspectives on the results. Our team holds the view that the result is not superficial. We believe the expected result is a triangular configuration. In order to illustrate this, we conducted an additional experiment incorporating mixup with the MSE loss. The resulting last-layer activations are arranged as convex combinations of the classifier, aligning with what we perceive as a superficial outcome. We hope that this additional experiment emphasizes the unexpected nature of the observed configuration.

---

> > ### Author Response · Authors · 2023-11-18
> > **Rebuttal by Authors 2/2**
> >
> > Now, we respond to the questions:
> >
> > 1. We are basing this claim on the test accuracies presented in Table 1. To improve the paper we have clarified this in the paper and mentioned that these accuracies are being compared to those found in other papers.
> >
> > 2. Though this is an interesting behaviour, and could potentially be investigated further, the authors would note that monotonicity is not necessarily a characteristic of Neural Collapse. Furthermore our measured values are comparable to that of the values found in the Neural Collapse paper. [2]
> >
> > 3. We were referring to the fact that, in Table 1, the Fashion MNIST mixup experiments saw the least improvement in test accuracy over the baseline. We have clarified this further in the paper following your question.
> >
> > 4. We have chosen to focus our investigation on why the geometric configuration is useful for explaining why mixup improves calibration. We suspect that the same geometric configuration could also explain why mixup improves generalization. As we mention in the paper, we suspect that the level of channel collapse is correlated with generalization. However, at this point we have no way of quantifying empirically or theoretically the level of channel collapse and therefore we leave this fascinating question for future work.
> >
> > 5. In this case, we are referring to the activations collapsing to the decision boundary. In hindsight, this phrasing is unclear and has since been changed.
> >
> > 6.  See response to weakness 2.
> >
> > We would like to reiterate our appreciation of the reviewer’s thorough and detailed response to our paper. We feel that the points raised and addressed have made our paper considerably stronger, and for that we thank you. Below we offer a summary of the overall changes made to the manuscript.
> >
> >
> > ### 1. **Enhanced Paper Structure and Readability**:
> >    - Figures are now strategically placed closer to the relevant text for a coherent reading experience.
> >    - Text from figure captions is integrated into the main body for better context.
> >    - Expanded the experiments section to provide more comprehensive insights.
> >
> > ### 2. **Additional Experiments, Analyses, and Figures**:
> >    - Updated the colors in Figure 1 for better visualization and added to it the classifiers.
> >    - Updated Figure 2, showcasing the feature configuration with MSE loss training, and its comparison with CE loss. This is meant to emphasize how the configuration obtained from the MSE loss is expected and uninteresting whereas the one obtained from CE, which is right next to it, is very different and unexpected.
> >    - Introduced Figure 3, illustrating the layer-wise trajectories of the CLS token for a ViT trained with mixup.
> >    - Introduced Table 2 reporting network calibration across various scenarios.
> >    - Introduced Figure 4 which elucidates the geometric configuration's role in improving calibration.
> >    - Introduced Figure 7 which presents changes in the mixup feature configuration once the last-layer classifier is fixed throughout training.
> >
> > ### 3. **Addressing Specific Reviewer Concerns**:
> >    - Added detailed explanation of 2D projection methods used in our visualizations.
> >    - Expanded on the theoretical framework, particularly regarding the assumptions about the simplex ETF classifier and its implications.
> >    - Investigated the layer-wise trajectory of representations and its implications in mixup training.
> >    - Discussed the practical significance of our findings in the context of calibration improvements.
> >
> > Through these revisions, we aim to bridge the gap between the practical success of mixup training and its theoretical understanding, providing a more comprehensive and valuable contribution to the field. In light of these refinements, should you find it fitting, we would be most grateful for any potential reconsideration of our score.
> >
> > [1] Hongyu Guo, Yongyi Mao, and Richong Zhang. Mixup as locally linear out-of-manifold regularization, 2018
> >
> > [2] Vardan Papyan, X. Y. Han, and David L. Donoho.
> > Prevalence of neural collapse during the terminal phase of deep learning training. Proceedings of the National Academy of Sciences, 117(40):24652–24663, sep 2020. doi: 10.1073/pnas.2015509117. URLhttps://doi.org/10.1073%2Fpnas.2015509117.

---

> > > ### Author Response · Authors · 2023-11-20
> > > **Rebuttal by Authors**
> > >
> > > Dear reviewer,
> > >
> > > As the rebuttal period is ending shortly, please let us know if you have any further questions or if we can provide further clarification.

---

> > > > ### Author Response · Authors · 2023-11-22
> > > > **Rebuttal by Authors**
> > > >
> > > > Dear Reviewer,
> > > >
> > > > Thank you for your valuable feedback on our manuscript. We have carefully addressed your comments and submitted a revised manuscript. We would greatly appreciate if you could take a moment to read our rebuttal before the deadline today. We understand the demanding nature of the review process and appreciate the time and effort you are dedicating to this task

---

> > > > > ### Comment · Reviewer_vgCZ · 2023-11-22
> > > > > **Response confirmation**
> > > > >
> > > > > Thanks for addressing my concerns with the detailed response and the thorough modifications on the paper. I have raised the rating from 5 to 6

---

### Official Review · Reviewer_BaMd · 2023-11-02

**Soundness:** 3 good
**Presentation:** 1 poor
**Contribution:** 2 fair
**Rating:** 6
**Confidence:** 4

**Summary:**

The study utilizes analytical methods from the Unconstrained Features Model (UFM) and Layer-Peeled Model (LPM) related to the Neural Collapse (NC) phenomenon to examine the mixup data augmentation technique. The paper provides an analytical solution illustrating feature alignment with class prototypes during same-class mixup and channel delineation along the decision boundary for different-class mixup. Empirical evidence is presented to support these analytical findings.

**Strengths:**

The strengths of the paper are evident, particularly in the analytical exploration of the Unconstrained Features Model's (UFM) dependency on the mixup technique. This analysis deepens the understanding of a standard practice and provides empirical confirmation of expected behaviors. The analytical depth of the study stands out as a significant contribution. Additionally, the detailed examination of the feature density for λ=0.5 adds a noteworthy dimension to the research.

**Weaknesses:**

- The organization of the paper could benefit from a more conventional structure. The introduction figures and the placement of experimental results in Section 2 present challenges to readability and could be restructured for clarity.

- A clearer presentation of the problem's structure, particularly how it has been addressed numerically, would enhance the reader's comprehension of the methodology and findings.

- While the analytical proof is a notable element of the paper, the practical significance of its findings could be better articulated to highlight their impact and possibly extend the manuscript's reach to a broader audience.

- The rationale for projecting features into 2D, as depicted in the introductory figure, is not immediately clear and warrants further explanation. Given the dimension-specific uniqueness of the regular d-Simplex ETF, further details on the projection process would be beneficial. It should be clarified whether the projection includes all classes or if any are excluded, and if so, whether this exclusion is systematic or random. Employing a pair plot or scatter plot matrix, as referenced in [1], may provide a more conventional representation of the multidimensional data and ensure clarity regarding the inclusion of all classes. Given that datasets such as CIFAR10 and FashionMNIST comprise 10 classes, they could be visualized directly without the need for projection.

[1] Pernici, F., Bruni, M., Baecchi, C., & Del Bimbo, A. (2019, June). Maximally Compact and Separated Features with Regular Polytope Networks. In CVPR Workshops (pp. 46-53).

- The part on 'amplification' related to decision boundaries would benefit from a more thorough explanation. This concept is central to the paper, and a detailed discussion is essential, extending beyond the brief description in Figure 4's caption. Providing clear evidence or a solid rationale for this phenomenon is crucial to enhance the paper's validity.To better understand the amplification effect, it may be helpful to initiate training with a regular d-simplex ETF that is fixed, as in [A] and [B]. By 'fixed,' it is meant that the parameters do not change during the training. This could provide clearer insights into the behavior of decision boundaries.

[A] Pernici, F., et al. (2021). Regular polytope networks. IEEE Transactions on Neural Networks and Learning Systems, 33(9), 4373-4387.

[B] Yang, Y., et al. (2022). Inducing Neural Collapse in Imbalanced Learning: Do We Really Need a Learnable Classifier at the End of Deep Neural Network?. Advances in Neural Information Processing Systems, 35, 37991-38002.

- Minor: Section 5.3 “additionally” seems to be a typo.

**Questions:**

Questions and weaknesses are grouped to provide a better understanding of the issues.

---

> ### Author Response · Authors · 2023-11-18
> **Rebuttal by Authors**
>
> We thank the reviewer for the time spent reading our paper and for your helpful and constructive review. Below we address the weaknesses and questions.
>
> - Upon your suggestion, we have revised the formatting of this paper and hope that it has improved its overall readability. Specifically, section 2 has been reworked with figures appearing closer to their relevant text sections.
> - We have re-written the problem statement of the paper and additional explanations for the numerical experiments have been incorporated, including a detailed explanation of the projection method.
> - We have included a clearer explanation of the interpretation of the theorem after the theorem statement.
> - We choose to project the features into 2D mainly to be consistent with the original Neural Collapse visualizations as well as other neural collapse-related papers and visualizations such as those cited in [1] - [4]. To address a more substantial set of classes, we have included a larger subset of classes in the appendix of the paper.
> - We have incorporated a more comprehensive exploration of the differences in the theoretical features in comparison to the empirical features. In response to the reviewer's suggestion, we conducted an experiment with the classifier fixed as a simplex ETF, contributing valuable insights to our analysis (as the resulting features align closer to the theoretical features). Additionally, we have included the details of the amplification used to generate Figure 6 in the Appendix.
>
>
> The authors would like to thank the reviewer for their thoughtful review as the changes implemented in response to the reviewer's comments have improved the paper's quality significantly. Below, we offer a summary of the overall changes to the manuscript.
>
> ### 1. **Enhanced Paper Structure and Readability**:
>    - Figures are now strategically placed closer to the relevant text for a coherent reading experience.
>    - Text from figure captions is integrated into the main body for better context.
>    - Expanded the experiments section to provide more comprehensive insights.
>
> ### 2. **Additional Experiments, Analyses, and Figures**:
>    - Updated the colors in Figure 1 for better visualization and added to it the classifiers.
>    - Updated Figure 2, showcasing the feature configuration with MSE loss training, and its comparison with CE loss. This is meant to emphasize how the configuration obtained from the MSE loss is expected and uninteresting whereas the one obtained from CE, which is right next to it, is very different and unexpected.
>    - Introduced Figure 3, illustrating the layer-wise trajectories of the CLS token for a ViT trained with mixup.
>    - Introduced Table 2 reporting network calibration across various scenarios.
>    - Introduced Figure 4 which elucidates the geometric configuration's role in improving calibration.
>    - Introduced Figure 7 which presents changes in the mixup feature configuration once the last-layer classifier is fixed throughout training.
>
> ### 3. **Addressing Specific Reviewer Concerns**:
>    - Added detailed explanation of 2D projection methods used in our visualizations.
>    - Expanded on the theoretical framework, particularly regarding the assumptions about the simplex ETF classifier and its implications.
>    - Investigated the layer-wise trajectory of representations and its implications in mixup training.
>    - Discussed the practical significance of our findings in the context of calibration improvements.
>
> Through these revisions, we aim to bridge the gap between the practical success of mixup training and its theoretical understanding, providing a more comprehensive and valuable contribution to the field. In light of these refinements, should you find it fitting, we would be most grateful for any potential reconsideration of our score.
>
> [1] Chenxi Huang, Liang Xie, Yibo Yang, Wenxiao Wang, Binbin Lin, & Deng Cai. (2023). Neural Collapse Inspired Federated Learning with Non-iid Data.
>
> [2] Rangamani, A., Lindegaard, M., Galanti, T. &amp; Poggio, T.A.. (2023). Feature learning in deep classifiers through Intermediate Neural Collapse. Proceedings of the 40th International Conference on Machine Learning, in Proceedings of Machine Learning Research 202:28729-28745 Available from https://proceedings.mlr.press/v202/rangamani23a.html.
>
> [3] Xie, L., Yang, Y., Cai, D., & He, X. (2022). Neural Collapse Inspired Attraction-Repulsion-Balanced Loss for Imbalanced Learning. Neurocomputing, 527, 60-70.
>
> [4] Yang, Y., Chen, S., Li, X., Xie, L., Lin, Z., & Tao, D. (2022). Inducing Neural Collapse in Imbalanced Learning: Do We Really Need a Learnable Classifier at the End of Deep Neural Network?. In Advances in Neural Information Processing Systems (pp. 37991–38002). Curran Associates, Inc..

---

> > ### Author Response · Authors · 2023-11-20
> > **Rebuttal by Authors**
> >
> > Dear reviewer,
> >
> > As the rebuttal period is ending shortly, please let us know if you have any further questions or if we can provide further clarification.

---

> > > ### Author Response · Authors · 2023-11-22
> > > **Rebuttal by Authors**
> > >
> > > Dear Reviewer,
> > >
> > > Thank you for your valuable feedback on our manuscript. We have carefully addressed your comments and submitted a revised manuscript. We would greatly appreciate if you could take a moment to read our rebuttal before the deadline today. We understand the demanding nature of the review process and appreciate the time and effort you are dedicating to this task

---

> > > > ### Author Response · Authors · 2023-11-23
> > > > **Rebuttal by Authors**
> > > >
> > > > As we near the end of the discussion period, we would like to highlight that 2/4 reviewers have raised their scores in response to our revisions. We have been eagerly anticipating your feedback, but unfortunately, we have not received a response yet. Nevertheless, we remain hopeful that our revisions have addressed the reviewer's concerns, and perhaps they will share their thoughts with the AC in the upcoming discussions.

---

### Official Review · Reviewer_cLyU · 2023-11-08

**Soundness:** 3 good
**Presentation:** 3 good
**Contribution:** 2 fair
**Rating:** 5
**Confidence:** 2

**Summary:**

This work explores the phenomenon of neural collapse for mix-up training strategy. Specifically, the authors show that mixup’s last-layer activations will converge to a distinctive configuration. Extensive visualization results are illustrated to demonstrate the conclusion.

**Strengths:**

1.	Rich visualization results verify the authors’ points.

**Weaknesses:**

1.	I am afraid that the contribution of this method is limited. The authors simply show that mixup training strategy will help mixup’s last-layer activations converge to a distinctive configuration. However, we usually more care about how to set a proper mixup rate $\lambda$ and which samples should we mixup.

**Questions:**

Please see the weakness.

---

> ### Author Response · Authors · 2023-11-18
> **Rebuttal by Authors**
>
> The authors would like to thank the reviewer for the time spent reading and reviewing our paper. To address the reviewer's point, about the contributions of the paper being limited, the authors have undertaken significant revisions to enhance the clarity, theoretical depth, and empirical evidence of our manuscript, by adding significantly more experimentation and analysis to the paper. Below is a summary of changes made to the manuscript:
>
> ### 1. **Enhanced Paper Structure and Readability**:
>    - Figures are now strategically placed closer to the relevant text for a coherent reading experience.
>    - Text from figure captions is integrated into the main body for better context.
>    - Expanded the experiments section to provide more comprehensive insights.
>
> ### 2. **Additional Experiments, Analyses, and Figures**:
>    - Updated the colors in Figure 1 for better visualization and added to it the classifiers.
>    - Updated Figure 2, showcasing the feature configuration with MSE loss training, and its comparison with CE loss. This is meant to emphasize how the configuration obtained from the MSE loss is expected and uninteresting whereas the one obtained from CE, which is right next to it, is very different and unexpected.
>    - Introduced Figure 3, illustrating the layer-wise trajectories of the CLS token for a ViT trained with mixup.
>    - Introduced Table 2 reporting network calibration across various scenarios.
>    - Introduced Figure 4 which elucidates the geometric configuration's role in improving calibration.
>    - Introduced Figure 7 which presents changes in the mixup feature configuration once the last-layer classifier is fixed throughout training.
>
> ### 3. **Addressing Specific Reviewer Concerns**:
>    - Added detailed explanation of 2D projection methods used in our visualizations.
>    - Expanded on the theoretical framework, particularly regarding the assumptions about the simplex ETF classifier and its implications.
>    - Investigated the layer-wise trajectory of representations and its implications in mixup training.
>    - Discussed the practical significance of our findings in the context of calibration improvements.
>
> Through these revisions, we aim to bridge the gap between the practical success of mixup training and its theoretical understanding, providing a more comprehensive and valuable contribution to the field. In light of these refinements, should you find it fitting, we would be most grateful for any potential reconsideration of our score.

---

> > ### Author Response · Authors · 2023-11-20
> > **Rebuttal by Authors**
> >
> > Dear reviewer,
> >
> > As the rebuttal period is ending shortly, please let us know if you have any further questions or if we can provide further clarification.

---

> > > ### Author Response · Authors · 2023-11-22
> > > **Rebuttal by Authors**
> > >
> > > Dear Reviewer,
> > >
> > > Thank you for your valuable feedback on our manuscript. We have carefully addressed your comments and submitted a revised manuscript. We would greatly appreciate if you could take a moment to read our rebuttal before the deadline today. We understand the demanding nature of the review process and appreciate the time and effort you are dedicating to this task

---

> > > > ### Author Response · Authors · 2023-11-23
> > > > **Rebuttal by Authors**
> > > >
> > > > As we near the end of the discussion period, we would like to highlight that 2/4 reviewers have raised their scores in response to our revisions. We have been eagerly anticipating your feedback, but unfortunately, we have not received a response yet. Nevertheless, we remain hopeful that our revisions have addressed the reviewer's concerns, and perhaps they will share their thoughts with the AC in the upcoming discussions.

---

### Meta-Review · Area_Chair_PRTZ · 2023-12-09

**Metareview:**

This study examines the mixup augmentation technique based on the neural collapse phenomenon and its analytical methods. Some important observations are discovered with theoretical explanations. This study brings interpretability of the mixup technique and through an analytical view from neural collapse. Most of the effective reviews suggest that this paper conducts a novel investigation, and provides adequate theoretical and experimental results. Some reviewers were concerned about the limited practical implications and unclear discussions, but acknowledged that their concerns were addressed in the authors’ response. The ACs agree with the reviewers and recommend accept. More thorough discussions and clear explanations w.r.t. the reviewers’ concerns should be included in the final paper.

**Justification For Why Not Higher Score:**

Limited practical implications and significances.

**Justification For Why Not Lower Score:**

The investigation is novel and interesting. Some important observations and theoretical insights that could deepen the understanding of mixup practice are provided.

---

### Decision · Program_Chairs · 2024-01-16

Accept (poster)